# Phase-Field Simulation of Microstructure Formation in Gas-Atomized Al–Cu–Li–Mg Powders

**DOI:** 10.3390/ma16041677

**Published:** 2023-02-17

**Authors:** May Pwint Phyu, Frank Adjei-Kyeremeh, Piyada Suwanpinij, Iris Raffeis, Markus Apel, Andreas Bührig-Polaczek

**Affiliations:** 1Foundry Institute, RWTH Aachen University, Intzestraße 5, 52072 Aachen, Germany; 2The Sirindhorn Thai-German Graduate School of Engineering (TGGS), King Mongkut’s University of Technology North Bangkok (KMUTNB), 1518 Pracharat 1 Road, Wongsawang, Bangsue, Bangkok 10800, Thailand; 3Leibniz-Institute für Werkstofforientierte Technologien-IWT, Badgasteiner Straße 3, 28359 Bremen, Germany; 4ACCESS e.V., Intzestraße 5, 52072 Aachen, Germany

**Keywords:** phase field simulation, Al–Cu, Al–Cu–Li, Al–Cu–Li–Mg, gas atomization, powder

## Abstract

Al–Cu–Li (2xxx series) powders for additive manufacturing processes are often produced by gas atomization, a rapid solidification process. The microstructural evolution of gas-atomized powder particles during solidification was investigated by phase-field simulations using the software tool MICRESS. The following topics were investigated: (1) the microsegregation of copper and lithium in the particle, and the impact of lithium addition on the formation of secondary phases in Al-2.63Cu and Al-2.63Cu-1.56Li systems, (2) the effect of magnesium on the nucleation and final mass fraction of T_1_ (Al_2_CuLi) growing from the melt, and (3) the effect of increased magnesium content on the T_1_ and Sʹ (AlCu_2_Mg) phase fractions. It is observed that the addition of lithium into the Al–Cu system leads to a decrease in the solid solubility of copper in the primary matrix; consequently, more copper atoms segregate in the interdendritic regions resulting in a greater mass fraction of secondary precipitates. Our result agrees with findings on the beneficial impact of magnesium on the nucleation and precipitation kinetics of T_1_ precipitates in the conventional casting process with further thermomechanical heat treatments. Moreover, it is observed that the increase in magnesium from 0.28 wt.% to 0.35 wt.% does not significantly affect the nucleation and the amount of the T_1_ phase, whereas a decrease in T_1_ phase fraction and a delay of T_1_ formation are encountered when magnesium content is further raised to 0.49 wt.%.

## 1. Introduction

Low-density, high-strength materials provide a high specific strength which is of great interest and ideal for engineering and structural applications. Pure aluminum (99% or higher) 1xxx series is a material relatively soft because of its atomic structure—a closed-packed FCC lattice with high stacking fault energy and a large number of slip systems which allows dislocation slipping even at low temperatures and does not display ductile to brittle transition property [1]. Therefore, with the help of alloying elements, many aluminum alloys have been developed with improved strength, density, etc., for structural applications.

Aluminum–lithium alloys [2,3,4,5] are well-known in the aerospace industry because of their high strength and lightweight properties. As the light metal lithium with the lowest density of the elements under stable standard conditions, the addition of lithium to aluminum alloys is advantageous. Each 1 wt.% up to 4 wt.% addition of lithium leads to a reduction in the density by 3% and an increase in the elastic modulus by approximately 6% [6]. Additions of Li together with Cu into an aluminum can form the major strengthening precipitate T_1_ (Al_2_CuLi) when the atom percentage of Cu is higher than that of Li. [5]. When the Li content is higher, the δ^′^ (Al_3_Li) phase may form. It is also a strengthening phase but leads to adverse effects, such as low anisotropic mechanical properties, poor corrosion resistance, low ductility, fracture toughness, and thermal instability arising from shearable δ^′^ precipitates. Other copper and lithium-bearing phases, such as T_B_ (Al_7.5_Cu_4_Li) and T_2_ (Al_6_CuLi_3_), may also form depending on the alloying compositional and thermomechanical treatments [7]. Magnesium is a beneficial alloying element in Al–Cu–Li alloys since it favors the nucleation of the T_1_ phase or promotes the formation of Sʹ (Al_2_CuMg) known from the ternary Al–Cu–Mg system. T_1_ has a hexagonal crystal structure showing a thin plate-like morphology aligned semi-coherently on {111} the matrix planes [8]. δ^′^ and T_B_ have cubic structures [7]. T_2_ also has a cubic structure with icosahedral symmetry, while the Sʹ phase is orthorhombic with a rod-like structure [4].

The competitive formation of the various potential phases in an Al–Cu–Li alloy is, among other factors, dependent on the alloy composition, processing technique, processing parameters, prior deformation by cold working, and subsequent artificial aging. Specifically, nucleation and precipitation of T_1_ are influenced by the lithium content [9,10], the Cu/Li ratio [11], minor alloying elements such as Mg and Ag [12,13,14], aging conditions, and prior deformation [15,16,17,18,19]. The abovementioned investigations are related to conventional processes and low cooling rates.

Unlike conventional casting processes, the nucleation conditions of T_1_ in Al–Cu–Li alloys in rapid solidification processes such as additive manufacturing and atomization processes are less understood. To shed more light on the microstructure formation at higher cooling rates, in this study, gas-atomized powders are investigated as an example. In a gas atomization process, thermal energy is extracted rapidly from dispersed, small molten metal droplets to the surrounding inert gas, leading to fast solidification rates. Typically, fast cooling of small droplets facilitates larger melt undercooling with a higher nucleation rate compared with conventional casting [20]. On the other hand, if the cooling rate is too high, the formation of some phases can be suppressed. T_1_ precipitates could not be detected in the samples manufactured by the laser power bed fusion process (LPBF) in the as-built (no preheating) condition [21]. This was the case even in areas that underwent the LPBF-typical remelting, resolidifying, and cyclic reheating with different cooling and heating rates. A study of the microstructure of additively manufactured Al–Cu–Li alloys LPBF under preheating conditions showed that prerequisites for the formation of T_1_ precipitates out of the melt and solid matrixes, in addition to the appropriate chemical composition, are suitable cooling rates for nucleation and heating regime for nuclei growth by diffusion [21,22].

Gas atomization is a complex process associated with two key mechanisms; (1) dispersion of the melt into small liquid metal droplets and (2) cooling and solidifying the droplets into powder particles. The spray process parameters also influence particle solidification, so many numerical models have been developed to understand the interaction mechanism between the atomizing gas and the melt, as well as the thermal physics of the powder atomization. Zeoli et al. [23] introduced a model in which both mechanisms of gas atomization (breakup and cooling) are combined. The authors stated that the initial droplet size is the influencing factor for the thermal history of the droplets. Smaller droplets have higher undercooling potential. Prasad et al. [24] developed a mathematical model to quantify the microsegregation of a system with a chemical composition able to form a eutectic phase at the end of solidification (weight percent eutectic) and solute solubility in the primary phase during rapid solidification of Al–Cu powders. The authors reported that the quantified microsegregation and solute solubility are not significantly influenced by the droplet size and gas type. Volloton et al. [25] presented a numerical model of eutectic growth coupled to different transient heat flow models for validation with experimental results from rapid solidification of Al-33 wt.%Cu droplets. The authors concluded that using Whitaker’s correlation with the thermophysical properties of atomizing gas gives the best approach for the heat transfer calculation of small and large droplets.

Phase-field modeling [26] is a tool to simulate the microstructure evolution, including microsegregation in a spatially and temporally resolved manner. It considers the thermal process history and is thus not bound to a global equilibrium constraint, i.e., it only assumes local equilibrium at the phase boundaries. By this, fast solidification processes can also be simulated. Phase-field modeling is more general compared with a Scheil–Gulliver calculation and includes the option to consider nucleation undercooling and kinetic effects or diffusion-limited growth kinetic. When linked to thermodynamic databases, phase-field models lead to quantitative predictions for the phase fractions and the spatial distribution of alloying elements. Therefore, they have a wide range of applications, e.g., simulation of phase transformations [27], heat treatments [28,29], gas atomization [30], additive manufacturing [31,32], and many more. A recent formulation of a multicomponent/multiphase field model, as it is, for instance, implemented in the software MICRESS, is given in the reference [33]. Phase-field models are also applicable for fast solidification, as shown in the references [34,35]. The link between the microstructure evolution governed by the phase field and diffusion equations and the temperature evolution based on the energy balance is described in more detail in [36].

Several investigations have been carried out by different authors focusing on the cooling rate estimation of gas-atomized Al–Cu powders using the dendrite arm spacing (DAS) measurement [37], microsegregation of Al–Cu alloy under different cooling rates using two-dimensional (2-D) pseudo-front tracking (PFT) model [38], and many more. No work has been carried out in the ternary system where Li is added into Al–Cu alloy, especially alloying element distribution and secondary phase formation in the field of droplet solidification using phase-field simulation. Understanding the microsegregation of Cu and Li together with nucleation and formation of secondary phases during solidification is important for subsequent process steps, such as heat treatment, as microsegregation affects the solid-state precipitation of T_1_, which contributes to strengthening mechanisms in Al–Cu–Li alloys. The aim of this phase-field simulation study was to investigate the solidification behavior of a gas-atomized droplet, including the formation of secondary phases, especially T_1_, with a special focus on the microsegregation of the light element Li. Experimentally, the Li distribution is difficult to measure as standard EDX-detectors cannot detect Li because of its low energy characteristics. Hence, a quantitative phase-field model has been applied with the further advantage of a higher spatial resolution compared with an SEM-EDX analysis of a single atomized droplet. The cooling curve, solidification path, the mass fraction of secondary phases, and the segregation of the alloying elements Cu and Li are analyzed and quantified by the phase-field simulations. The effect of adding lithium on the microsegregation of copper and secondary phase fraction is investigated by comparing Al-2.63Cu and Al-2.63Cu-1.56Li powders. In addition, various magnesium contents were added to Al-2.63Cu-1.56Li to study its impact on the T_1_ nucleation and phase fraction. Furthermore, simulation results were compared with results from the literature, and our own experimental results were compared with a commercially purchased Al–Cu–Li–Mg powder.

## 2. Materials and Methods

### 2.1. Material

In this study, commercially purchased gas-atomized Al–Cu–Li–Mg powder from Nanonval GmbH (Berlin, Germany) was used for the purpose of validating the simulation results. The nominal composition of the powder is listed in Table 1.

### 2.2. Methodology

#### 2.2.1. Simulation Model

In this paper, the phase-field simulation of the gas-atomized powder was performed using MICRESS software (Version 6.303, ACCESS e.V., Aachen, Germany) [39]. The simulations for the binary alloy Al-2.63Cu considered two solid phases, primary FCC_Al and θʹ (Al_2_Cu). Subsequently, the model was extended to multicomponent alloy systems with lithium and magnesium. For the alloy Al-2.63Cu-1.56Li, the formation of three solid phases, such as primary FCC_Al, T_1_, and T_B_ (Al_7.5_Cu_4_Li), were taken into consideration. The droplet solidification of the alloy Al-2.63Cu-1.56Li-xMg comprises the formation of FCC_Al, T_1_, and Sʹ. In the phase-field simulation scenario, only the cooling process was considered since the aim of the present study was to understand the microsegregation and phase formation in the solidifying droplet. Table 2 summarizes the input parameters that were used in the droplet solidification model of gas-atomized powder. The initial concentrations of the alloying elements, i.e., copper, lithium, and magnesium, used in different simulations of Al–Cu, Al–Cu–Li, and Al–Cu–Li–Mg are described in Table 3.

It was required to define the phase interaction between the liquid and all solid phases, the primary FCC phase, and all other secondary phases, e.g., T_1_, T_B_, and Sʹ phase, in order to enable the growth or shrinkage of one phase at the expense of the other. The main condition for the nucleation of secondary phases from the melt was a critical nucleation undercooling, set to 3K for all secondary phases in all simulations. In addition, a nucleation temperature range, i.e., the minimum and maximum temperature, was defined for each desired second phase in which the precipitation of these second phases was expected. The nucleation ranges for the secondary precipitates in this model were taken from Thermocalc Scheil calculation; 750 and 821 °C for θʹ in Al-2.63Cu, 730 and 813 °C, 730 and 798 °C for T_1_ and T_B_, respectively, in Al-2.63Cu-1.56Li, and 700 and 786 °C for Sʹ phase in Al-2.63Cu-1.56Li-0.28Mg. The maximum number of possible nuclei was not limited, but a minimal distance (shield distance) of 1µm was introduced to avoid the growth of new nuclei of the same phase around the initial nucleus due to the release of latent heat.

The grid spacing was set to ∆x = 0.02 µm, which was a compromise between resolution and calculation time in such a way that changes in grid spacing no longer significantly affected the simulation results. A two-dimensional simulation domain (circle shaped) was used instead of a three-dimensional spherical model as it offers results with sufficient accuracy, e.g., high spatial resolution with reasonably short calculation times. Only a quarter of a circle was used for simulation because of the symmetrical characteristic of the droplet, as demonstrated in Figure 1. There were a total of 504 cells in the X and Z directions, and the number of cells in the Y direction was set to 1 as it is a 2D simulation. The thickness of the diffuse interface was set to 0.08 µm (four grid cells).

The phase-field equation was solved together with an energy balance equation for the temperature evolution and diffusion equations for the concentration field of the solute atoms. The phase-field model, i.e., the computation of driving forces for interface motion and nucleation and solute partitioning at the moving solid–liquid interfaces, was linked to the thermodynamic database TTAl8. The mobility database MOBAL1 along with the thermodynamic database was used to compute the diffusion coefficients of solute atoms in the liquid and primary solid FCC phases. The diffusion coefficient values described in Table 2 represent global equilibrium values which are temperature- and composition-dependent in the specific phases.

In this simulation scenario, the temperature evolution and, thus, the cooling rate were determined from the energy balance between external cooling, heat capacity, and latent heat. The cooling of the droplet was determined by a heat extraction rate (HER) (J/s*cm^2^) as a measure of the heat transfer from the droplet to the surrounding inert gas by convection and radiation. However, the contribution of heat loss caused by radiation was very small, while convection was mainly responsible for droplet solidification. The thermal gradient was neglected as it was assumed that heat conductivity within the droplet is fast.

#### 2.2.2. Approach

This research focused on the formation of eutectic fractions and microsegregation of alloying elements during the solidification of gas-atomized powder particles. A fast cooling rate was achieved when the liquid droplet was exposed to the inert gas in the atomization chamber. Consequently, mixing and redistributing alloying elements may differ from conventional casting with lower cooling rates, and the eutectic fraction may vary due to lower nucleation temperatures of the secondary phase.

In this research, the droplet solidification of Al-2.63Cu during gas atomization was simulated first, and then it was extended to Al-2.63Cu-1.56Li and Al-2.63Cu-1.56Li-0.28Mg. In addition, the magnesium content was varied in Al-2.63Cu-1.56Li-xMg. The comparison was made between Al-2.63Cu and Al-2.63Cu-1.56Li alloy solidification in terms of the cooling curve, solidification path, eutectic fraction, and segregation of copper atoms during solidification. Then, the influence of magnesium on the T_1_ phase precipitation was studied. The last investigation was carried out to observe how varying magnesium contents in the quaternary alloy affects the mass fraction of the T_1_ and Sʹ phase.

## 3. Results and Discussion

### 3.1. Phase Field Simulation

#### 3.1.1. Al-2.63Cu and Al-2.63Cu-1.56Li

The precipitation sequence of Al–Cu–Li alloys exhibited features of both binary Al– Cu and Al–Li systems [11]. In Al–Cu, it was accompanied by the formation of θʹ precipitates and δʹ (Al_3_Li) phase in binary Al–Li alloys. However, T_1_ was the major strengthening precipitate in Al–Cu–Li alloys. Its formation, particularly solid-state precipitation, depended on many factors such as Cu/Li ratio, minor alloying elements, or prior deformation by cold working. Therefore, to design an optimal heat treatment process, it was important to understand the nucleation and growth of all Cu- and Li-containing phases already nucleating from the melt along the solidification path, particularly in rapid solidification processing.

The simulated cooling curve for the heat extraction rate given in Table 2 and the corresponding solid fraction formation of both Al-2.63Cu and Al-2.63Cu-1.56Li are presented in Figure 2a,b, respectively. It can be seen in Figure 2a that addition of 1.56 wt.% of lithium into Al-2.63Cu led only to a slight decrease in the onset of solidification by nucleation of the primary FCC phase from 920 °C to 918 °C. The eutectic reaction also started at a lower temperature, 805 °C for Al-2.63Cu-1.56Li, compared with 815 °C for Al-2.63Cu. Additionally, the total amount of secondary phases in Al-2.63Cu-1.56Li was increased from approximately 2 to 2.5 wt.% compared with Al-2.63Cu, as shown in Figure 3a. There was only one secondary phase, θʹ (Al_2_Cu), in Al-2.63Cu, whereas two secondary phases were formed in Al-2.63Cu-1.56Li alloy, namely T_1_ and T_B_ (see Figure 3b). The decrease in nucleation temperature of the secondary phases when adding Li to the binary Al-2.63Cu can be seen directly in Figure 3b.

Huang and Zheng [9] reported that for conventional casting with additional aging treatment, θʹ phase formation is observed to be less frequent in the range of Li concentration from 0.5 wt.% to 1.6 wt.% and is entirely suppressed at a high Li level of 1.6 wt.%. In order to verify the effect of lithium concentration on secondary phase formation during rapid solidification, Scheil calculations were carried out, and the results are presented in Figure 4. According to the Scheil results, it may be concluded that θʹ formation can be expected when the lithium content is lower than 0.5 wt.%; above this, the θʹ formation is compensated by T_B_ and T_1_ formation during rapid solidification.

During solidification, the partitioning of lithium, copper, and other alloying elements at the growing solid–liquid interface leads to an inhomogeneous element distribution in the solidified particle. Post-heat treatment, such as homogenization, can eliminate these microsegregation effects [40]. However, the general understanding of the segregation in the as-solidified material is important for the adjustment of subsequent processing steps.

As an example, Al–Cu with different copper contents was simulated. The copper pileup ahead of the interface increased with increasing Cu concentration in the alloy, see Figure 5. The snapshots in Figure 5a–c were taken at the same time, showing the effect of the Cu content on the growth dynamics as well. The red color in Figure 5a–c indicates the solute pileup regions with the highest copper concentration (e.g., approximately 3 wt.% Cu in Al-1.5Cu, 5 wt.% Cu in Al-2.63Cu, and 8 wt.% Cu in Al-4.5Cu), while dark blue and light blue represent FCC region and melt (1.5 wt.% Cu in Al-1.5Cu, 2.63 wt.% Cu in Al-2.63Cu, and 4.5 wt.% Cu in Al-4.5Cu), respectively. The copper concentration profiles in Figure 5a–c are also represented by the graph in Figure 5d. The graph in Figure 5e gives a sorted representation of the Cu concentration per computation grid cell, a common method to analyze the microsegregation [41,42]. In the simulation settings, the position of the FCC nucleus is predefined in the lower left corner of the simulation domain to respect the symmetry. In reality, nucleation is expected to take place at the surface rather than in the center. However, for a statistical investigation of the segregation, the position was not that relevant; more important were the length scales, i.e., dendrite arm spacing and cooling rate. As it is later discussed, the length scales of the simulation and experiment are comparable. The outward-growing dendrite leads to a Cu accumulation between the main branches (interdendritic region) and at the particle surface, which then exhibits the eutectic areas. In Figure 5e, the graph shows that copper concentration dissolves in the primary FCC matrix in Al-1.5Cu is 0.3 wt.%. It further increases to 0.5 wt.% and 0.8 wt.% with increasing nominal copper content in Al-2.63Cu and Al-4.5Cu, respectively.

In Figure 6, a comparison between Al-2.63Cu and Al-2.63Cu-1.56Li is shown. The addition of Li as a second alloying element affected the partitioning of Cu at the solidification front, i.e., the effective distribution coefficient for Cu, c^Cu^_sol_/c^Cu^_liq_, is smaller, and the amount of copper dissolved in the primary FCC phase has dropped from approximately 0.5 wt.% to 0.3 wt.%. Consequently, the amount of copper pushed into the remaining liquid was larger for the Al-2.63Cu-1.56Li alloy compared with Al-2.63Cu, which led to a higher amount of secondary phases, i.e., fraction eutectic.

The image in Figure 7 shows the local Cu composition in the completely solidified powder particle. The segregation analysis in Figure 7b confirms a higher Cu concentration in the FCC phase in the binary alloy Al-2.63Cu compared with the ternary Al-2.63Cu-1.56Li alloy, although the nominal Cu concentration was the same for both alloys. It is reported that the solid solubility of copper decreased when lithium atoms were added [7]. Consequently, there is a larger amount of copper available for the formation of secondary phases.

The segregation of lithium in Al-2.63Cu-1.56Li followed the same trend as Cu, but the segregation was not as strong. Approximately 1.3 wt.% of Li was dissolved in the FCC matrix, as represented in Figure 8. The segregated copper together with lithium led to the formation of ternary phases, such as T_1_ (Al_2_CuLi) and T_B_ (Al_15_Cu_8_Li_2_), in the interdendritic regions. In Al-2.63Cu, the amount of copper needed to form θʹ in the interdendritic areas was larger. Therefore, the fraction of secondary phases was smaller, as shown in Figure 3a.

When taking a closer look at the concentration gradient of lithium atoms in Figure 8a, a depletion of lithium (dark blue spots) can be found between the secondary phases in the interdendritic regions and the FCC matrix.

#### 3.1.2. Al-2.63Cu-1.56Li and Al-2.63Cu-1.56Li-0.28Mg

During solidification, T_1_, T_B_, and Sʹ precipitates grow partially out of the melt. For the solid-state precipitation of T_1_ at lower temperatures, a significant influence of magnesium is known from conventionally produced Al–Cu–Li alloys. It is reported in reference [12] that T_1_ precipitates heterogeneously and nucleates on dislocations and grain boundaries, and the addition of Mg and Ag leads to a uniform dispersion of T_1_ plates in the matrix of the conventionally produced Al–Cu–Li alloys. Moreover, it is expected that the addition of Mg to Al–Cu–Li alloys either favors the T_1_ nucleation or promotes the formation of the Sʹ phase, which belongs to the ternary Al–Cu–Mg system (see Scheil calculation in Figure 9).

To study the effect of Mg on the solidification behavior, phase field simulations with Al-2.63Cu-1.56Li-0.28Mg were performed in the identical simulation setting as before. Figure 10 presents the cooling curve and solid-fraction formation. The addition of 0.28 wt.% Mg did not significantly affect the melting temperature, nucleation temperature of the primary phase, solid fraction, and solidification range, unlike the lithium addition into Al-2.63Cu alloy. T_1_ temperature and fraction formation over time are presented in Figure 11a,b, respectively, showing that the nucleation temperature is not affected by magnesium addition. However, the T_1_ solid fraction increased slightly from 0.025% to 0.028%.

#### 3.1.3. Al-2.63Cu-1.56Li with Different Mg Contents: 0.28 wt.%, 0.35 wt.%, and 0.49 wt.%

To further investigate the effect of Mg on the phase formation during solidification, different amounts of Mg were investigated (0.28 wt.%, 0.35 wt.%, and 0.49 wt.%). The overall amount of the second phase fraction increases slightly (see Figure 12a). It can be seen in Figure 12b,c that when the initial composition of magnesium increased from 0.28 wt.% to 0.35 wt.%, the amount of T_1_ remained constant at approximately 2.87% and T_1_ started to appear around the same time (at 0.00686 s) during solidification. The amount of the Sʹ phase for the alloy with 0.35 wt.% magnesium (represented by the pink line in Figure 12c) was 0.1%, which is greater than that for the alloy with 0.28 wt.% magnesium. When the magnesium content was further increased up to 0.49 wt.%, it had an adverse effect on the T_1_ precipitation with a decrease in the T_1_ solid fraction, while the Sʹ phase fraction increased. Moreover, the T_1_ precipitation was delayed by approximately 0.0004 s compared with the alloy with 0.28 wt.% and 0.48 wt.% magnesium.

### 3.2. Experimental Results

#### 3.2.1. Powder Characterization by EDX Analysis

To analyze the segregation of the alloying elements (Cu and Mg) in the interdendritic regions, energy dispersive X-ray analysis (EDX) was performed. Here, EDX was used to determine the chemical composition of the material at specific points of the powder particles. Lithium, a light element, has a low energy characteristic X-ray; therefore, detecting Li in the standard EDX analysis was impossible.

The specific points for the elemental analysis for Al-2.63Cu-1.56Li-0.28Mg powder particles are represented in Figure 13. The results from the measurements of each spectrum of Al-4.5Cu and Al-2.63Cu-1.56Li-0.28Mg powder particles are given in weight percent and listed in Table 4.

According to the results from EDX point analysis, the solute concentration of both Cu and Mg was higher in the interdendritic phases, as expected from the segregation analysis (Figure 7) during phase-field simulation. In spectra 7, 8, 9, 13, and 14, which are the points in the interdendritic regions, the measured values for Cu (in weight percent) were 14.3, 11.8, 8, 7, and 4.6, respectively. The values were comparatively higher than those measured in the primary dendritic FCC phase, whose values were represented by spectra 10, 11, and 12. Similarly, the segregation of Mg in the interdendritic phases was found to be higher compared with the FCC matrix phase.

When comparing the results from the EDX point analysis and the segregation analysis from the phase-field simulation, it is observed that the amount of copper in the primary matrix and interdendritic region detected from the EDX point measurement was lower than that of the segregation analysis from the phase-field simulation. For instance, a maximum value of 14.3 wt.% of copper was detected in spectrum 7 in Table 4, while the copper concentration rose to 50 wt.% according to the results from phase-field simulation. We attribute this discrepancy to the limited spatial resolution of a “local” EDX point analysis. In the measurement, the concentration at a specific point is averaged over the volume where the electron beam generates the characteristic X-ray emission. For example, if the measured EDX point is focused on the interdendritic region with a higher concentration of segregated alloying elements, it will also include the FCC phase, which surrounds the interdendritic region due to the small length scale of the particle solidification microstructure. Therefore, the results from local EDX measurement are an average value over a larger volume than one captured by the results from the phase-field simulation determined by the grid spacing, which was 0.02µm in our case. It is also worth mentioning that in the phase-field simulations, concentration values were smeared over diffuse-interface regions. However, a single grid cell outside the interface has the “true” phase concentration. Hence, the effect of the diffuse interfaces can be seen in the steepness of the concentration distribution but not in the min/max values themselves (refer to Figure 7). To further confirm the explanation for the difference between EDX measurements and phase-field simulations, the simulated concentration field was mapped on different coarser grids. Figure 14 shows the sorted Cu concentration of the simulation results obtained for Al-2.63Cu-1.56Li plotted for different grid resolutions. It is evident that an evaluation on a coarser grid led to Cu concentrations that were closer to the local EDX measurement results.

#### 3.2.2. Eutectic Fraction Measurement Using Image Analysis

Figure 15 compares the solidified microstructure from the simulation and a cross-section SEM micrograph. It shows that the results from simulation and experiments are comparable in terms of the size of the segregation. Unlike in the simulated microstructure, subtle changes in copper concentration in the primary matrix and interdendritic regions could not be detected even using the backscattered electron (BSE) mode in the SEM image.

Backscattered electron (BSE) images of the gas-atomized Al-2.63Cu-1.56Li-0.28Mg powders (see Figure 16a–c) were also used in order to determine the amount of the eutectic fraction. Three measurements were carried out using three different particles of the same diameter (20 µm) with the help of the threshold feature from the ImageJ software. The results from the measurement are summarized in Table 5.

The total phase fraction of secondary phases in the 20 µm diameter Al-2.63Cu-1.56Li-0.28Mg particle obtained from the phase-field simulation was 2.98%, see Table 5. The secondary phases, T_1_, T_B_, and Sʹ, in the solidified Al-2.63Cu-1.56Li-0.28Mg droplet, could not be discriminated by EDX or image analysis, but the amount of the eutectic fraction as estimated by the threshold analysis compared well with the simulation results. As the eutectic also comprised the FCC phase, which was not counted in the mass fraction of secondary phases from the phase-field simulations, the eutectic fraction was underestimated in the simulation. A more detailed and quantitative analysis of the experimental data, together with simulation results, is subject to future work.

## 4. Conclusions

The solidification behavior of fast-solidifying Al–Cu–Li–Mg melt droplets during gas atomization was studied with the help of phase-field simulations using the software MICRESS. Previous discussions of microsegregation and phase formation during solidification of Al–Cu–Li alloys, especially T_1_ precipitation and eutectic fraction, are often based on the literature results obtained for conventional casting, i.e., slow solidification rates. Only a few studies were found for higher cooling rates. Reflecting on the results from the investigations conducted in this study, the following conclusions can be made:(i)The results from 2D phase-field simulations of solidifying droplets of different Al–Cu–Li–Mg alloys, radius r = 10 µm, with heat extraction rates expected for gas atomization, are consistent with experimental SEM and EDX investigations;(ii)The simulations allow a quantitative analysis of the element distribution and phase fractions of the as-solidified droplet beyond the resolution limit of the experimental SEM-EDX analysis. Other analytical methods need to be used to validate the simulation;(iii)In particular, for the FCC matrix phase, the amount of dissolved Li and Cu can be determined by the simulations. Moreover, the as-solidified phase fractions of θʹ, T_1_, T_B_, and Sʹ can be simulated, depending on the alloy composition. The simulation results show that:(iv)The addition of lithium to a binary Al–Cu alloy decreases the amount of dissolved copper in the primary FCC matrix;(v)The total amount of secondary phases in Al-2.63Cu-1.56Li is larger than in Al-2.63Cu, not only because of the additional alloying element lithium but also because of more copper segregating into the interdendritic regions, forming T_1_ and T_B_ precipitates together with Li;(vi)It is shown that the addition of magnesium of up to 0.35 wt.% has a significant effect on the T_1_ precipitation leading to an increase in T_1_ fraction. However, a further increase in magnesium content of up to 0.49 wt.% leads to a delay in T_1_ formation, a decrease in T_1_ fraction, and an increase in Sʹ fraction.

## Figures and Tables

**Figure 1 materials-16-01677-f001:**
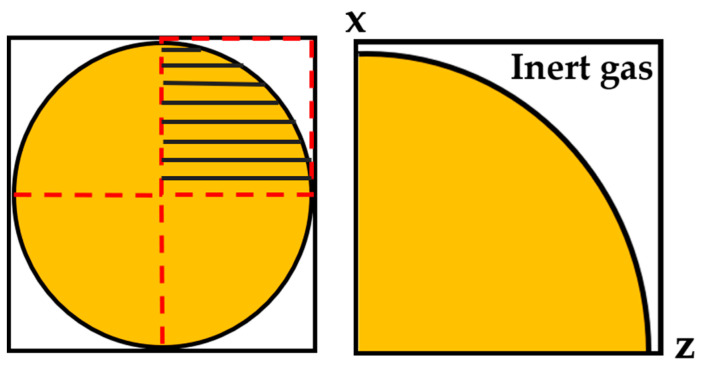
Two-dimensional geometric model of droplet solidification.

**Figure 2 materials-16-01677-f002:**
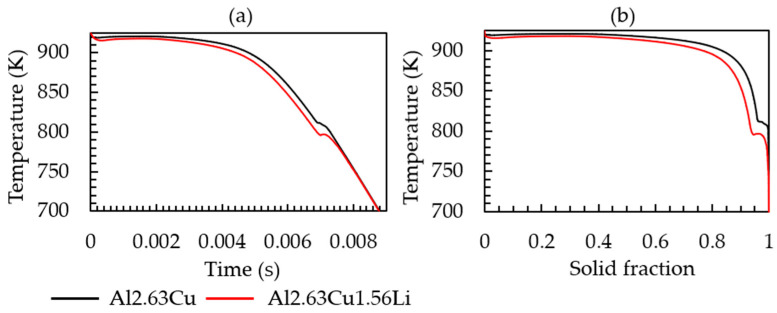
(**a**) Time-temperature cooling curve; (**b**) solid fraction-temperature curve of the Al-2.63Cu and Al-2.63Cu-1.56Li alloys for a constant heat extraction rate of 1.53 × 10^5^ Js^−1^cm^−3^.

**Figure 3 materials-16-01677-f003:**
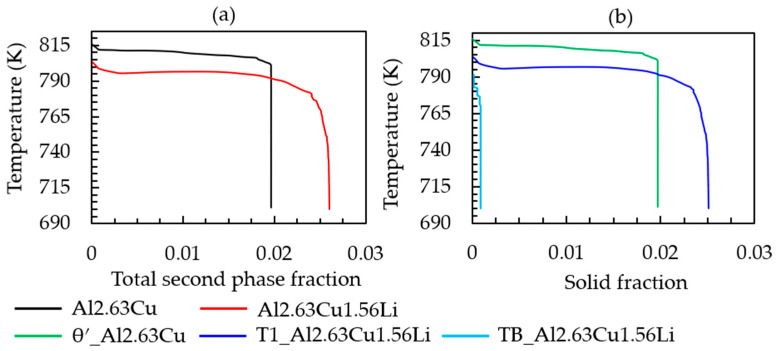
(**a**) Temperature versus total second phase fraction; (**b**) amount of θʹ, T_1_, and T_B_ fraction versus time of Al-2.63Cu and Al-2.63Cu-1.56Li droplet solidification.

**Figure 4 materials-16-01677-f004:**
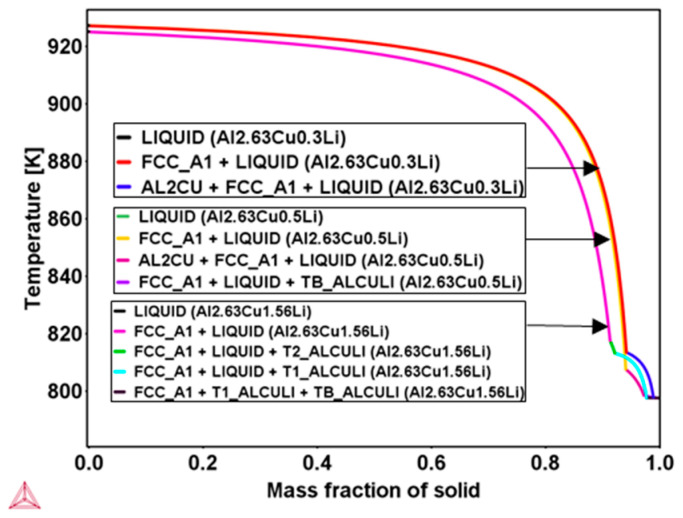
Scheil simulation using the database TTAl8 in Thermo-Calc software for three different Al-2.63Cu-xLi alloys, Li = 0.3 wt.%, 0.5 wt.%, and 1.56 wt.%, showing the solidification path, primary FCC, and secondary phases, indicating no θʹ is found in the Al2.63Cu0.5Li and Al2.63Cu1.56Li.

**Figure 5 materials-16-01677-f005:**
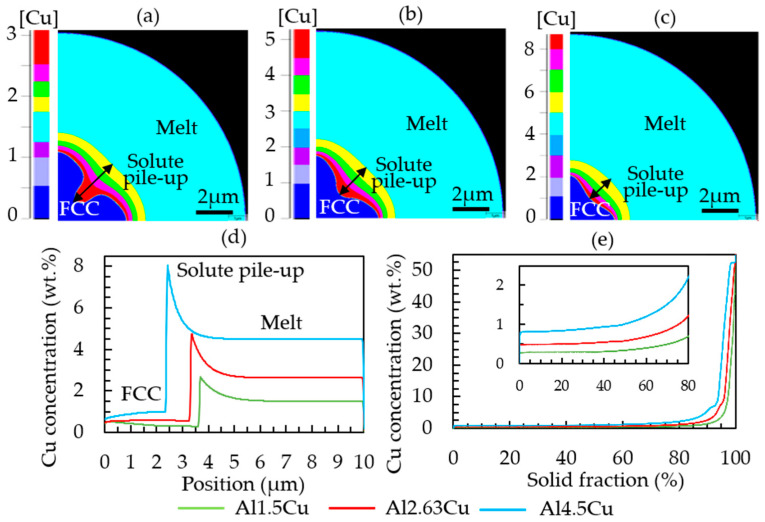
Diagram showing the copper pileup (in wt.%) ahead of the solid–liquid interface after 0.0004 s of growth: (**a**) Al-1.5Cu, (**b**) Al-2.63Cu, (**c**) Al-4.5Cu, (**d**) 1D plot of the Cu concentration along the Z direction from cell 1 to 500, and (**e**) sorted local copper concentration values in the completely solidified Al–Cu droplets.

**Figure 6 materials-16-01677-f006:**
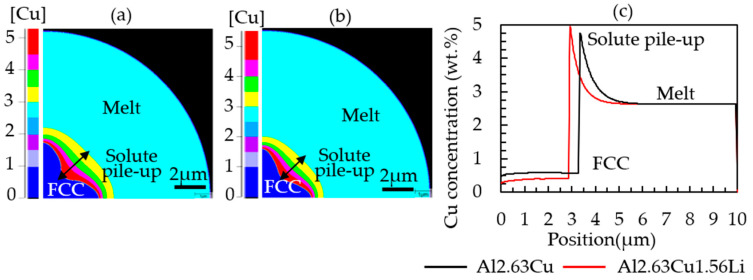
Diagram showing the copper pileup (in wt.%) ahead of the solid–liquid interface at simulation time 0.0004 s during droplet solidification of (**a**) Al-2.63Cu and (**b**) Al-2.63Cu-1.56Li, and (**c**) comparison along the Z direction from cell 1 to 500.

**Figure 7 materials-16-01677-f007:**
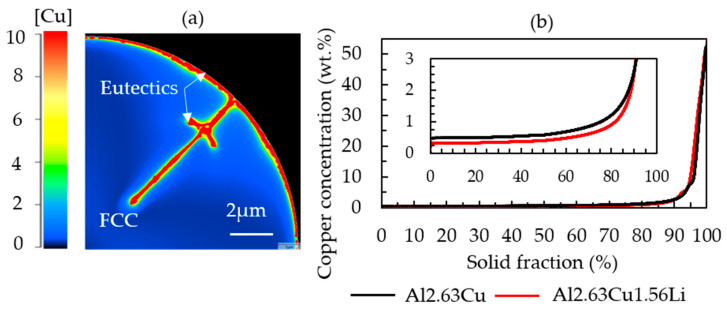
(**a**) Simulated microstructure of a completely solidified Al-2.63Cu-1.56Li powder showing the copper distribution. (**b**) Sorted local copper concentration values in Al-2.63Cu-1.56Li and Al-2.63Cu.

**Figure 8 materials-16-01677-f008:**
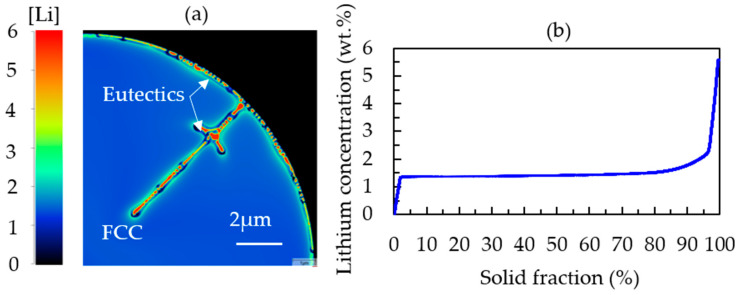
(**a**) Simulated microstructure of a completely solidified Al-2.63Cu-1.56Li particle showing the lithium distribution. (**b**) Sorted local lithium concentration values in Al-2.63Cu-1.56Li.

**Figure 9 materials-16-01677-f009:**
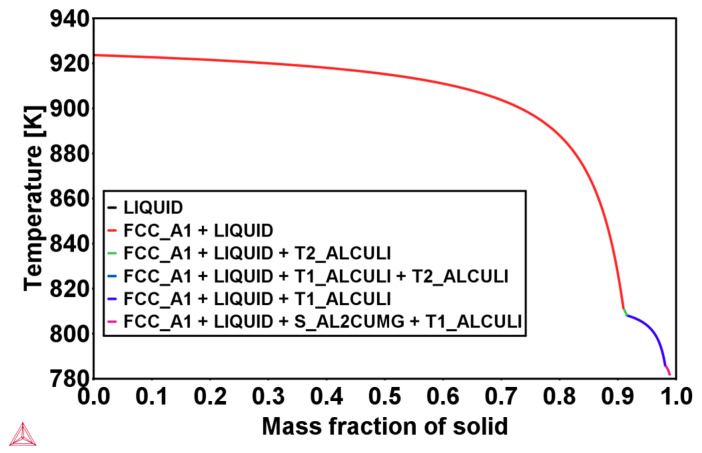
Scheil calculation using the database TTAl8 in Thermo-Calc software for Al-2.63Cu-1.56Li-0.28Mg.

**Figure 10 materials-16-01677-f010:**
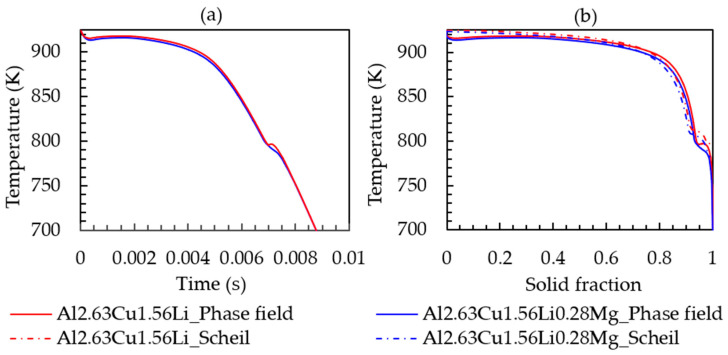
(**a**) Time-temperature cooling curve; (**b**) solid fraction-temperature curve of Al-2.63Cu-1.56Li and Al-2.63Cu-1.56Li-0.28Mg droplet solidification.

**Figure 11 materials-16-01677-f011:**
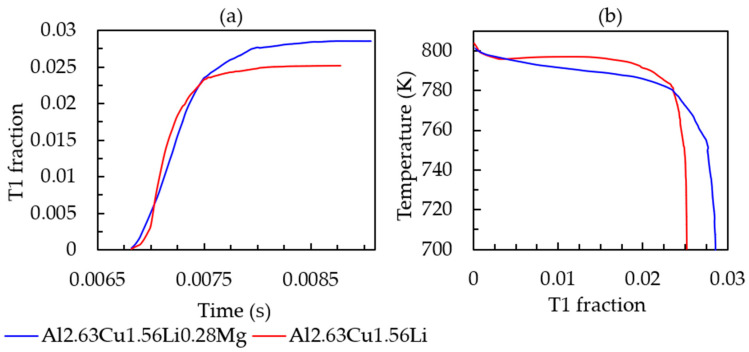
(**a**) T_1_ fraction versus simulation time. (**b**) T_1_ fraction versus temperature for Al-2.63Cu-1.56Li and Al-2.63Cu-1.56Li-0.28Mg droplet solidification.

**Figure 12 materials-16-01677-f012:**
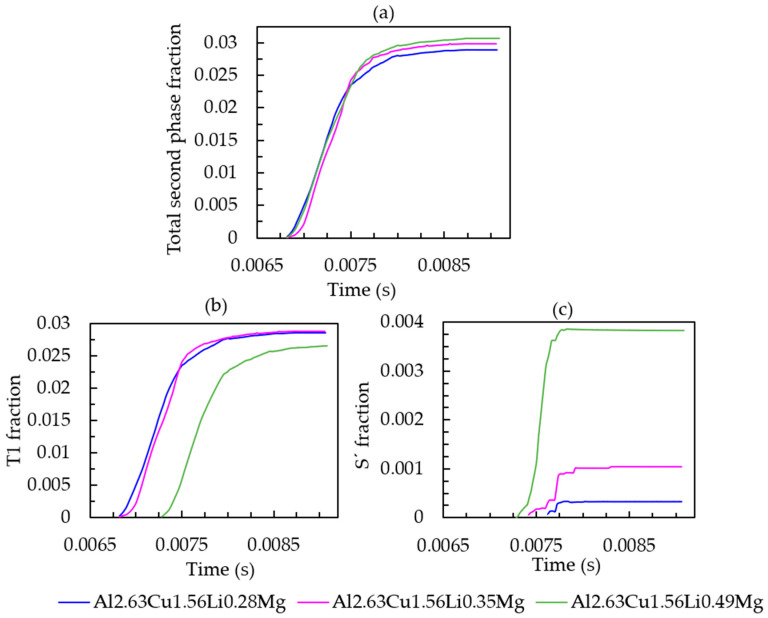
(**a**) Total second phase fraction versus simulation time; (**b**) T_1_ fraction versus simulation time; (**c**) Sʹ fraction versus simulation time of Al-2.63Cu-1.56Li-xMg.

**Figure 13 materials-16-01677-f013:**
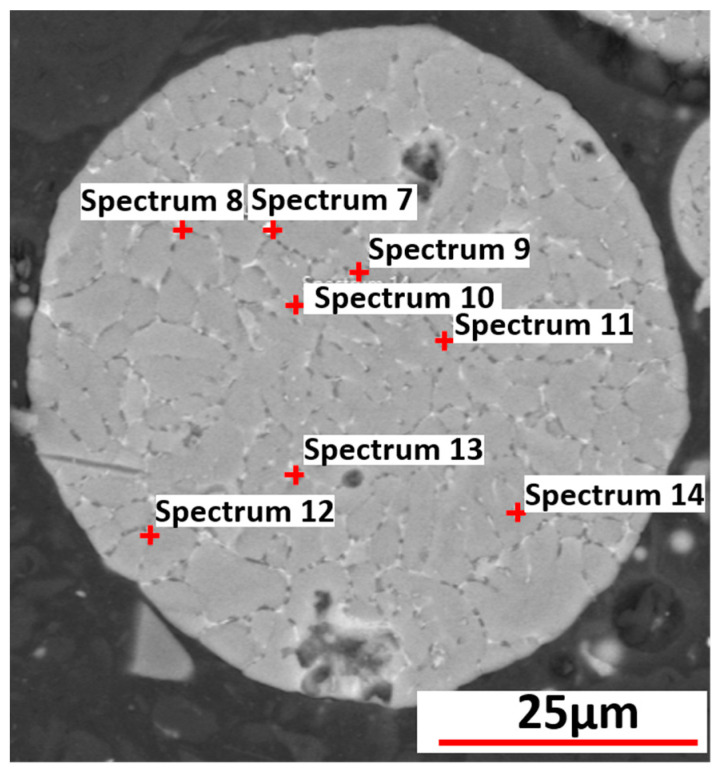
EDX analysis of Al-2.63Cu-1.56Li-0.28Mg powder.

**Figure 14 materials-16-01677-f014:**
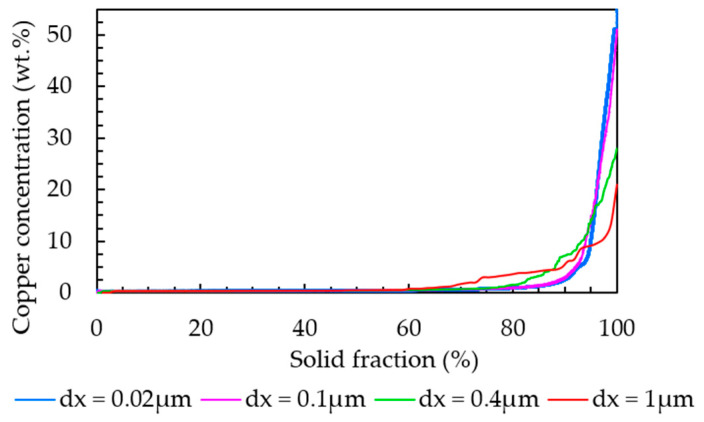
Sorted copper concentration values from phase-field simulations for an Al-2.63Cu-1.56Li droplet evaluated with different grid resolutions.

**Figure 15 materials-16-01677-f015:**
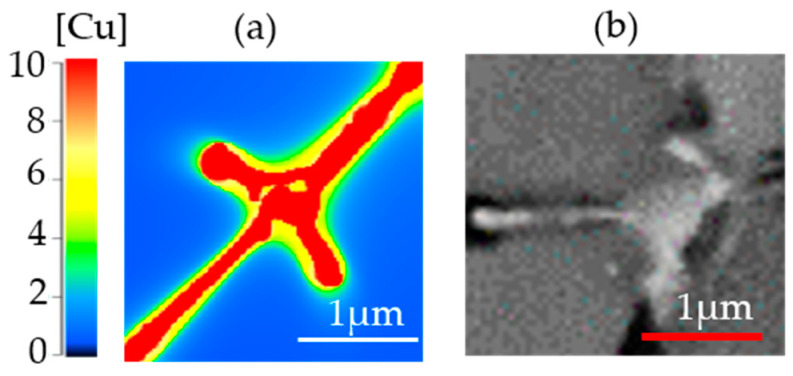
(**a**) Simulated microstructure and (**b**) cropped micrograph of Al-2.63Cu-1.56Li-0.28Mg powder.

**Figure 16 materials-16-01677-f016:**
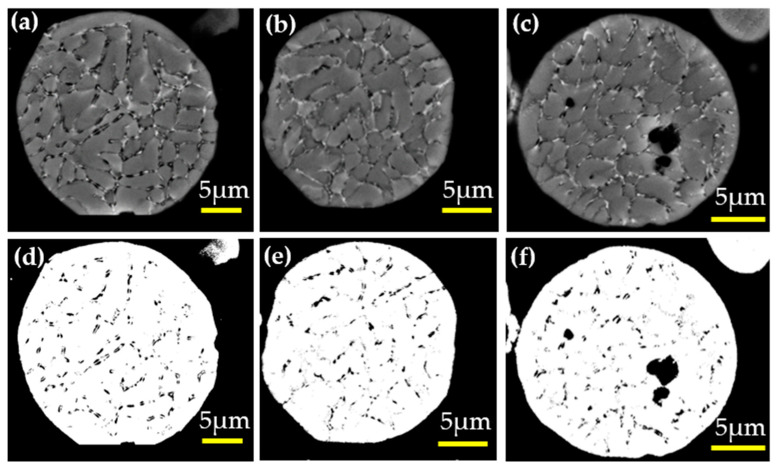
(**a**–**c**) Gas-atomized 20 µm Al-2.63Cu-1.56Li-0.28Mg powder in backscattered electron primary FCC phase (gray) and interdendritic phases (black); (**d**–**f**) Eutectic areas of Al-2.63Cu-1.56Li-0.28Mg powder by a threshold algorithm using ImageJ.

**Table 1 materials-16-01677-t001:** The nominal composition of gas-atomized Al–Cu–Li–Mg powder.

Material	Nominal Composition (wt.%)							
Al–Cu–Li–Mg	Al	Cu	Li	Mg	Zn	Mn	Zr	Si
	Bal	2.63	1.56	0.28	0.67	0.17	0.09	0.02

**Table 2 materials-16-01677-t002:** Input parameters for the phase-field simulation.

Input Parameters		
Initial conditions	Domain size (number of cells)	X = 504, Y = 1, Z = 504
	Droplet geometry	Droplet—round shape and 10 µm in radius surrounded by an
		Inert phase
	Initial temperature at the bottom	925 K
Process parameters	Boundary conditions	ssss (s = symmetrical)
	Heat extraction rate (HER)	1.53 × 10^5^ J/s*cm^3^
Material parameters	Thermodynamic database	TTAl8
	Mobility database	MOBAL1
	Interfacial energy	Solid phases/melt = 2.45 × 10^−5^ J/cm^2^
		Solid–solid interfaces = 2.45 × 10^−5^ J/cm^2^
	Diffusion coefficients	Cu in melt = 4.66 × 10^−5^ cm^2^/s
		Li in melt = 2.01 × 10^−5^ cm^2^/s
		Mg in melt = 8.85 × 10^−5^ cm^2^/s
		Cu in FCC phase = 1.1 × 10^−8^ cm^2^/s
		Li in FCC phase = 2.52 × 10^−8^ cm^2^/s
		Mg in FCC phase = 2.04 × 10^−8^ cm^2^/s
Numerical parameters	Grid spacing	0.02 µm
	Interfacial thickness	4 cells (0.08 µm)

**Table 3 materials-16-01677-t003:** The initial concentration used for Al–Cu, Al–Cu–Li, and Al–Cu–Li–Mg droplets solidification.

Initial Concentration	
Al-2.63Cu	Cu—2.63 wt.%
	Al—Balance
Al-2.63Cu-1.56Li	Cu—2.63 wt.%
	Li—1.56 wt.%
	Al—Balance
Al-2.63Cu-1.56Li-0.28Mg	Cu—2.63 wt.%
	Li—1.56 wt.%
	Mg—0.28 wt.%
	Al—Balance
Al-2.63Cu-1.56Li-xMg	Cu—2.63 wt.%
	Li—1.56 wt.%
	Mg—0.28 wt.%, 0.35 wt.% and 0.49 wt.%
	Al—Balance

**Table 4 materials-16-01677-t004:** EDX results of Al-2.63Cu-1.56Li-0.28Mg powder.

Al-2.63Cu-1.56Li-0.28Mg Powder					
Spectrum	Cu in wt.%	Mg in wt.%	Spectrum	Cu in wt.%	Mg in wt.%
7	14.3	0.3	11	2.4	0.2
8	11.8	0.4	12	2.7	0.2
9	8	0.3	13	7	0.4
10	1.8	0.1	14	4.6	0.4

**Table 5 materials-16-01677-t005:** Eutectic fraction measurement of gas-atomized 20µm Al-2.63Cu-1.56Li-0.28Mg powder.

	Al-2.63Cu-1.56Li-0.28Mg Powder
Phase field simulation	2.98%
Measurement using image analysis	3.69%

## Data Availability

Not applicable.

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
