# Peer review of "Phase-Field Simulation of Microstructure Formation in Gas-Atomized Al–Cu–Li–Mg Powders"

_materials, 2023, doi:10.3390/ma16041677_

Round 1

Reviewer 1 Report

This manuscript investigates the microstructural evolution of Al-Cu-Li-Mg gas-atomized powder particles during solidification, by phase field simulations, using the software tool MICRESS. The authors investigate and draw conclusions for the micro-segregation of Cu and Li in 10um 2-D particles, and effect of Mg content on T1 and S’ fractions.

The manuscript is in good shape and the subject matter is of importance. However, based on the methodology, results, and discussion sections of the manuscript, reviewer has the following questions and suggestions:

-       Noting that a better understanding of nucleation and phase fractions of fast solidification processes is beneficial, I have a hard time understanding the connection and arguments regarding additively manufactured Al-Cu-Li alloys made by the authors. Specifically:

“It is indicated that T1 precipitates are not detected in the as-built condition which matches the finding regarding the difficulty of T1 nucleation without prior deformation or heat treatment during the conventional process.”

Is it hypothesized that the existence of T1 phase in powder particles would persist within additively manufactured components after undergoing complex temporal heat processing? In my opinion, a better clarification of this introduction section would be beneficial, not to add too much more to the already lengthy introduction section.

-       As the authors mentioned and considering the limitations of SEM-EDX method, an experimental and quantitative analysis might be due in the future to validate these simulations. In the opinion of the reviewer, this fact undermines the second conclusion made by the authors, stating:

“The simulations allow a quantitative analysis of the element distribution and phase fractions of the as solidified droplet beyond the resolution limit of the experimental analysis.

-       In the opinion of the reviewer, there are multiple sections that require proper referencing such as:

“Additions of Li together with Cu into aluminum can form the major strengthening precipitate, T1 (Al2CuLi) when the atom percentage of Cu is higher than that of Li.

And

“Typically, fast cooling of small droplets facilitates larger melt undercooling with a higher nucleation rate compared to conventional casting.”

And most importantly:

“During solidification, the partitioning of lithium, copper and other alloying elements at the growing solid/liquid interface leads to an inhomogeneous element distribution in the solidified particle. Post-heat treatment such as homogenization can eliminate these microsegregation effects.”

Which seems to be speculative.

Author Response

Dear Reviewer,

Minor revisions requested by you can be seen in 

Line 51

Line 70-86

Line 267

Line 460-461

This manuscript investigates the microstructural evolution of Al-Cu-Li-Mg gas-atomized powder particles during solidification, by phase field simulations, using the software tool MICRESS. The authors investigate and draw conclusions for the micro-segregation of Cu and Li in 10um 2-D particles, and effect of Mg content on T1 and S’ fractions.

The manuscript is in good shape and the subject matter is of importance. However, based on the methodology, results, and discussion sections of the manuscript, reviewer has the following questions and suggestions:

  • Noting that a better understanding of nucleation and phase fractions of fast solidification processes is beneficial, I have a hard time understanding the connection and arguments regarding additively manufactured Al-Cu-Li alloys made by the authors. Specifically:

“It is indicated that T1 precipitates are not detected in the as-built condition which matches the finding regarding the difficulty of T1 nucleation without prior deformation or heat treatment during the conventional process.”

Is it hypothesized that the existence of T1 phase in powder particles would persist within additively manufactured components after undergoing complex temporal heat processing? In my opinion, a better clarification of this introduction section would be beneficial, not to add too much more to the already lengthy introduction section.

Unlike conventional casting processes, the nucleation conditions of T1 in Al-Cu-Li alloys in rapid solidification processes such as additive manufacturing and atomization processes are less understood. To shed more light on the microstructure formation at higher cooling rates, in this study, gas atomized powders are investigated as an example. In a gas atomization process, thermal energy is extracted rapidly from dispersed, small molten metal droplets to the surrounding inert gas leading to fast solidification rates. Typically, fast cooling of small droplets facilitates larger melt undercooling with a higher nucleation rate compared to conventional casting [20]. On the other hand, if the cooling rate is too high, the formation of some phases can be suppressed. T1 precipitates could not be detected in the samples the laser power bed fusion process (LPBF) in the as-built (no preheating) condition [21]. This was the case even in areas which underwent the LPBF-typical remelting, resolidifying and reheating with different cooling and heating rates. A study of the microstructure of additively manufactured Al-Cu-Li alloys LPBF under preheating conditions showed that prerequisites for the formation of T1 precipitates out of the melt and solid matrixes, in addition to the appropriate chemical composition, are suitable cooling rates for nucleation and a heating regime for nuclei growth by diffusion. [21, 22]. (Changes can be seen in line 70, 71, 72 and 77-86.)

(2)        As the authors mentioned and considering the limitations of SEM-EDX method, an experimental and quantitative analysis might be due in the future to validate these simulations. In the opinion of the reviewer, this fact undermines the second conclusion made by the authors, stating:

“The simulations allow a quantitative analysis of the element distribution and phase fractions of the as solidified droplet beyond the resolution limit of the experimental analysis.

The simulations allow a quantitative analysis of the element distribution and phase fractions of the as solidified droplet beyond the resolution limit of the experimental SEM-EDX analysis.  Other analytical methods need to be used to validate the simulation. (Changes in line (Added in line 460-461)

  • In the opinion of the reviewer, there are multiple sections that require proper referencing such as:

“Additions of Li together with Cu into aluminum can form the major strengthening precipitate, T1 (Al2CuLi) when the atom percentage of Cu is higher than that of Li.” (N. E. Prasad, A. A. Gokhale, and R. J. H. Wanhill, Eds., Aluminum-lithium alloys: processing, properties, and applications. Oxford: Elsevier, Butterworth-Heinemann, 2014.) (Added in line 51)

And

“Typically, fast cooling of small droplets facilitates larger melt undercooling with a higher nucleation rate compared to conventional casting.” (M. E. F. Fine, Rapidly Solidified Powder Aluminum Alloys: A Symposium. ASTM International, 1986.) (Added in line 77)

And most importantly:

“During solidification, the partitioning of lithium, copper and other alloying elements at the growing solid/liquid interface leads to an inhomogeneous element distribution in the solidified particle. Post-heat treatment such as homogenization can eliminate these microsegregation effects.” (D. W. Heard, J. Boselli, R. Rioja, E. A. Marquis, R. Gauvin, and M. Brochu, ‘Interfacial morphology development and solute trapping behavior during rapid solidification of an Al–Li–Cu alloy’, Acta Materialia, vol. 61, no. 5, pp. 1571–1580, Mar. 2013, doi: 10.1016/j.actamat.2012.11.034) (Added in line 267)

Reviewer 2 Report

Dear Authors

Unfortunately, the lines in the manuscript are not numbered, making reviewing more complex. Please, connect the uncommented highlighted text fragments with the below-listed questions and suggestions

Please, in the introduction, add a sentence explaining why Your study is required, as far as later, You say that Your results are in agreement with the others. By what Your study differs from the others?

Do You know the solidification rate of the droplets in Your experiment? This is important to compare results (part 2.1)

In table 4, please, add the data about all the spectra from Fig. 13

Figure 15. The simulated microstructure significantly differs from that obtained by SEM. Please, explain this

Author Response

Dear Reviewer,

Minor revisions requested by you can be seen in 

Line 122-131

Line 170-171

Line 255-257

Line 422-425

(1)         Please, in the introduction, add a sentence explaining why Your study is required, as far as later, you say that Your results are in agreement with the others. By what Your study differs from the others?

Added in line 122-131

Several investigations have been carried out by different authors focusing on the cooling rate estimation of gas atomized Al-Cu powders using the dendrite arm spacing (DAS) measurement [37], microsegregation of Al-Cu alloys under different cooling rate using two-dimensional (2-D) pseudo-front tracking (PFT) model [38] and many more. No work has been done in the ternary system where Li is added into Al-Cu alloy especially alloying element distribution and secondary phase formation in the field of droplet solidification using phase field simulation. Understanding the microsegregation of Cu and Li together with nucleation and formation of secondary phases during solidification is important for subsequent process steps like heat treatment, as microsegregation affects the solid state precipitation of T1, which contributes to strengthening mechanisms in Al-Cu-Li alloys.

(2)         Do You know the cooling rate of the droplets in Your experiment? This is important to compare results (part 2.1)

We did the dendrite arm spacing (DAS) measurement using the intersect method in order to estimate the cooling rate of the droplets in our powder. The DAS measurement is done according to ref: Eli Vandersluis, Comondore Ravindran, ‘Comparison of Measurement Methods for Secondary Dendrite Arm Spacing’, Metallogr. Microstruct. Anal. (2017), Vol.6, Pp.89-94.) and  cooling rate is estimated  using the ref: G. Kasperovich, T. Volkmann, L. Ratke, and D. Herlach, ‘Microsegregation during Solidification of an Al-Cu Binary Alloy at Largely Different Cooling Rates (0.01 to 20,000 K/s): Modeling and Experimental Study’, Metall and Mat Trans A, vol. 39, no. 5, pp. 1183–1191, May 2008, doi: 10.1007/s11661-008-9505-6.).  From the DAS measurement, the cooling rate is approximately 1.8 x 104 K/s. On the other hand, the cooling rate calculated using the output data from the phase field simulation is approx. 1.3-1.7 x 104 K/s. Therefore, the cooling rate estimated from the DAS measurement and that from the simulation is comparable.

 (3)        Why it is important to define the phase interaction?

It is required to define the phase interaction between the liquid and all solid phases, the primary FCC phase and all other secondary phases, e. g. T1, TB and S‘ phase in order to enable the growth or shrinkage of one phase at the expense of the other. (Added in line 170-171)

(4)         Add reference: Already, Scheil calculations verify the effect of lithium during solidification: Li additions favors the formation of TB and T1 at the expense of θÍ´ (Al2Cu) when Li is increased up to 0.5wt.%

Changes are done in Line 255-257.

(5)         In table 4, please, add the data about all the spectra from Fig. 13

We here included the EDS results as a supplementary file so that you can kindly check out the spectra data we have put inside the manuscript. Please feel free to check it out.

(6)         Figure 15. The simulated microstructure significantly differs from that obtained by SEM. Please, explain this.

It shows that the results from simulation and experiments are comparable in terms of the size of the segregation. Unlike in the simulated microstructure, fine changes in copper concentration in the primary matrix and interdendritic regions cannot be detected even using the back scattered electron (BSE) mode in the SEM image.  (Added in line 422-425)

Reviewer 3 Report

1. Clearly mention the research gap before the aim

2. Include the node/elements used for this simulation

3. Improve the figure quality used for this investigation

4. Discussion part must me improved with recently published articles

Author Response

Dear Reviewer,

Minor revisions requested by you can be seen in 

Line 122-131

Line 188-189

  1. Clearly mention the research gap before the aim

Added in line 122-131

Several investigations have been carried out by different authors focusing on the cooling rate estimation of gas atomized Al-Cu powders using the dendrite arm spacing (DAS) measurement [37], microsegregation of Al-Cu alloys under different cooling rate using two-dimensional (2-D) pseudo-front tracking (PFT) model [38] and many more. No work has been done in the ternary system where Li is added into Al-Cu alloy especially alloying element distribution and secondary phase formation in the field of droplet solidification using phase field simulation. Understanding the microsegregation of Cu and Li together with nucleation and formation of secondary phases during solidification is important for subsequent process steps like heat treatment, as microsegregation affects the solid-state precipitation of T1, which contributes to strengthening mechanisms in Al-Cu-Li alloys.

  1. Include the node/elements used for this simulation

Added in line 188-189

In this simulation, elements are known as cells. There are total 504 cells in X and Y directions and number of cells in Z direction is set as 1 as it is 2D simulation.

  1. Improve the figure quality used for this investigation
  2. Discussion part must me improved with recently published articles

Dear Reviewer,

I tried hard to look for the recently published articles which we can use as citations. However, most of the articles I have found so far regarding this Al-Cu-Li alloys are only related to the solid-state transformation during different processes such as ageing, laser rapid melting and welding using laser metal deposition.

The following are the articles I have found.

[1]         S. M. Arbo, S. Tomovic-Petrovic, J. Aunemo, N. Dahle, and O. Jensrud, ‘On weldability of aerospace grade Al-Cu-Li alloy AA2065 by wire-feed laser metal deposition’, Journal of Advanced Joining Processes, vol. 5, p. 100096, Jun. 2022, doi: 10.1016/j.jajp.2022.100096.

[2]         X. Lei et al., ‘Investigation on laser beam remelted Al–Cu–Li alloy part I: Segregation and aging behavior at grain and dendrite boundaries’, Journal of Alloys and Compounds, vol. 855, p. 157519, Feb. 2021, doi: 10.1016/j.jallcom.2020.157519.

[3]         Z. Sun, X. Tian, B. He, Z. Li, and H. Tang, ‘Microstructure evolution and microhardness of the novel Al–Cu–Li-xSc alloys fabricated by laser rapid melting’, Vacuum, vol. 189, p. 110235, Jul. 2021, doi: 10.1016/j.vacuum.2021.110235.

[4]         X. Zhang et al., ‘Laser welding introduced segregation and its influence on the corrosion behaviour of Al-Cu-Li alloy’, Corrosion Science, vol. 135, pp. 177–191, May 2018, doi: 10.1016/j.corsci.2018.02.044.